# Defective Integrator activity shapes the transcriptome of patients with multiple sclerosis

Yevheniia Porozhan[1,*], Mikkel Carstensen[2,4,*], Sandrine Thouroude[1], Mickael Costallat[1], Christophe Rachez[1], Eric Batsché[1], Thor Petersen[3,†], Tove Christensen[2,†], Christian Muchardt[1]

**HP1α/CBX5 is an epigenetic regulator with a suspected role in multiple sclerosis (MS). Here, using high-depth RNA sequencing on monocytes, we identified a subset of MS patients with reduced CBX5 expression, correlating with progressive stages of the disease and extensive transcriptomic alterations. Examination of rare non-coding RNA species in these patients revealed impaired maturation/degradation of U snRNAs and enhancer RNAs, indicative of reduced activity of the Integrator, a complex with suspected links to increased MS risk. At protein-coding genes, compromised Integrator activity manifested in reduced pre-mRNA splicing efficiency and altered expression of genes regulated by RNA polymerase II pause–release. Inactivation of Cbx5 in the mouse mirrored most of these transcriptional defects and resulted in hypersensitivity to experimental autoimmune encephalomyelitis. Collectively, our observations suggested a major contribution of the Integrator complex in safeguarding against transcriptional anomalies characteristic of MS, with HP1α/CBX5 emerging as an unexpected regulator of this complex's activity. These findings bring novel insights into the transcriptional aspects of MS and provide potential new criteria for patient stratification.**

## Introduction

Multiple sclerosis (MS) is an acquired demyelinating disorder of the central nervous system (CNS) characterized by chronic inflammation and the formation of scar tissue (sclerosis) in various areas of the CNS. It is the most common disabling neurological disease of young adults, and currently available treatments primarily focus on managing the symptoms.

During the course of the disease, immune cells breach the blood–brain barrier and attack the myelin sheath that insulates neurons. This leads to symptoms such as numbness in various body parts, paresis, coordination and balance difficulties, blurred vision, slurred speech, and cognitive changes. MS can manifest in several distinct forms. Clinically isolated syndrome represents the initial clinical manifestation of a condition with features of inflammatory demyelination, which may suggest the possibility of MS, but does not yet fulfill the criteria for dissemination in time and space required for a definitive diagnosis. Once criteria for dissemination in time are met, the most common form of MS is relapsing–remitting MS (RRMS), characterized by clearly defined relapses of symptoms followed by periods of partial or complete recovery. RRMS may evolve into secondary progressive MS, where the disease worsens gradually without distinct relapses or remissions. Another form, primary progressive MS (PPMS), is characterized by a gradual worsening of symptoms from the onset, without distinct relapses or remissions (1).

The etiology of the disease remains unclear and is a subject of ongoing debate. Generally, it is believed that CNS-directed autoimmunity in MS results from a complex interplay of genetic susceptibility, hormonal factors, and environmental cues. These environmental cues include lifestyle aspects such as exposure to tobacco smoke and organic solvents, EBV infection, obesity during adolescence, and limited sun exposure coupled with low vitamin D levels. Some of these factors, particularly EBV serology, obesity, and possibly vitamin D deficiency, seem to be particularly relevant when they occur during adolescence (2). This early exposure contrasts with the typical onset of multiple sclerosis, which often does not manifest until the second or third decade of life, hinting at the existence of a long-term memory mechanism within the immune system. Although the modulation of the peripheral adaptive immune response has been extensively studied in this regard (3), the role of epigenetic alterations in creating a lasting imprint on the immune cells' transcriptional activity represents an underexplored but potentially significant memory mechanism, bridging the gap between early-life environmental exposures and later disease onset.

[1]Institut de Biologie Paris-Seine (IBPS), CNRS UMR 8256, Biological Adaptation and Ageing, Sorbonne Université, Paris, France   [2]Department of Biomedicine, Aarhus University, Aarhus, Denmark   [3]Department of Neurology, Hospital of Southern Jutland and Research Unit in Neurology, Department of Regional Health Research, University of Southern Denmark, Odense, Denmark   [4]Department of Clinical Medicine, Aarhus University, Forum, Aarhus, Denmark

Correspondence: christian.muchardt@sorbonne-universite.fr
*Yevheniia Porozhan and Mikkel Carstensen contributed equally to this work
†Thor Petersen and Tove Christensen contributed equally to this work

Previously, we have explored this possibility by examining the activity of HP1α/CBX5, a transcriptional repressor with affinity for both chromatin and RNA (4, 5, 6), and encoded by a gene sharing one of its promoters with *hnRNPA1*, a gene harboring single nucleotide polymorphisms (SNPs) previously associated with an elevated risk of MS (7). Although CBX5 is best known for its affinity for the heterochromatin-enriched histone H3 lysine 9 trimethylation (H3K9me3) mark, it is also active in euchromatin (8, 9). Accordingly, we showed that CBX5 was involved in repressing a set of cytokine genes and that in patients with MS, its recruitment to the promoters of these genes was reduced. This reduced recruitment could at least in part be correlated with increased citrullination of histone H3 at arginine 8 (H3Cit8), destroying the CBX5 binding site on the histone tail (10). Thus, reduced CBX5 activity in patients with MS offers a potential rationale for the concurrent elevation in the activity of proinflammatory cytokine genes and normally heterochromatinized DNA repeats, such as endogenous retroviruses (HERVs).

In a more recent study, we have further shown that at least a fraction of the HERV transcripts produced in MS patients are enhancer RNAs (eRNAs) originating from viral sequences coopted as immune-gene enhancers, and reactivated in the patients (11). Thus, CBX5 may also function as a regulator of eRNA production. Aligning with such a function, other HP1 family proteins are known to participate in RNA metabolism; notably, SWI6 is responsible for the degradation of heterochromatic transcripts in yeast (12), and HP1γ (CBX3) influences the regulation of alternative splicing in mammals (13, 14, 15, 16, 17). We therefore wished to re-explore the role of CBX5 in MS in light of high-depth genome-wide RNA-seq data allowing for detection of rare RNA species such as eRNAs and alternative splice junctions. This analysis has revealed that reduced levels of HP1α/CBX5 in monocytes from MS patients correlate with a range of transcriptional abnormalities all indicative of diminished Integrator complex (INTcom) activity. Therefore, the INTcom, crucial for non-coding RNA (ncRNA) maturation and RNA polymerase II (RNAPII) elongation, emerges as a novel player in MS pathology, offering mechanistic insights into numerous gene deregulation events associated with the disease.

# Results

### A subset of MS patients displays extensive transcriptional deregulation in monocytes

In an earlier study, we had observed elevated levels of HERV-encoded epitopes in MS patient monocytes (18). This cell type therefore appeared as well suited for the exploration of unusual RNA transcripts specific to the disease. With the objective of obtaining high-quality RNA-seq allowing detection of such rare RNA species, we collected monocytes from a cohort of 18 MS patients and seven control patients (Fig 1A). To explore eventual changes in the RNA populations induced by the disease progression, we selected a very heterogeneous set including patients with familial MS (#68, #125, and #130—one family, three generations), under immune-modulating treatment (#125, #127, #135, #136, and #139), or

with comorbidities (psoriasis, IDDM, Graves' disease, Crohn's disease, and ulcerative colitis). The control group, designated as symptomatic controls (SCs), comprised individuals who had sought medical assistance for symptoms suggestive of MS, but who, upon examination, exhibited no objective clinical or paraclinical findings conclusive for a specific neurological disease at the time of sample collection (Fig 1A). We can therefore state with high confidence that these control donors are free of MS. The RNA-seq analysis was conducted using a stranded paired-end protocol, yielding an output of ~110 million reads per sample. Principal component analysis on the RNA-seq data, profiling the 500 most variable genes, identified a substantial proportion of variance in gene expression across the samples. It further revealed the existence of two distinct biological subgroups, with PC1 scores that were either lower or significantly higher than those of the SCs (Fig 1B and Table S1A–C).

We further noted that expression levels of CBX5 varied among patients and that ranking the patients according to their respective CBX5 expression levels essentially recapitulated the principal component analysis. Indeed, patients with CBX5 expression levels in the lower quartile (Lo-CBX5) largely coincided with the patients strongly affected in their transcriptome, whereas patients with CBX5 expression levels in the upper quartile (Hi-CBX5) remained transcriptionally close to the SCs (Fig 1A–C). The highly transcriptionally divergent/Lo-CBX5 patients were all either RRMS, secondary progressive MS, or PPMS patients, whereas the Hi-CBX5 group was enriched in early-stage patients (Fig 1D). None of the other criteria proved relevant (patient age, gender, or comorbidities).

Pathway analysis on the genes differentially expressed between Hi-CBX5 patients and SCs (38 up-regulated genes and 109 down-regulated genes, twofold or more, adj. $P < 0.05$, baseMean>10—Fig 1E) revealed that down-regulated genes were enriched in genes regulated by the vitamin D receptor (analysis of the ChEA database, Fig 1F). This observation may reflect the importance of vitamin D in early phases of the disease.

In contrast to the Hi-CBX5 patients, Lo-CBX5 patients displayed a dramatically high number of genes with modified expression when compared to the SCs, with a total of 1,395 up-regulated and 1,350 down-regulated genes (twofold or more, adj. $P < 0.05$, baseMean>10—Fig 1G). As anticipated from earlier studies, pathway analysis using the KEGG database on up-regulated genes identified activation of the IL-17, TGF-beta, NFkappaB, and MAPK pathways, as well as genes associated with hypoxia (19) (Fig S1A). The pathway analysis also detected significant enrichment in genes associated with ferroptosis (Fig S1A, line 4). This form of regulated cell death characterized by the iron-dependent accumulation of lipid peroxides has previously been linked to neurodegeneration in MS (20). It is induced either through inactivation of the glutathione peroxidase GPX4, the enzyme responsible for detoxifying lipid hydroperoxides, or by functional inhibition of the cystine–glutamate antiporter, a two-subunit complex composed of SLC7A11 and SLC3A2. Yet, in the monocytes of the Lo-CBX5 patients, although p53 (TP53), which exerts a protective role against ferroptosis, was strongly down-regulated (twofold, adj. $P < 10^{-4}$), we observed only a moderate up-regulation of enzymes promoting lipid peroxidation (ACSL4 and LPCAT3). In addition, the detoxifying pathways (SLC7A11, SLC3A2, and GPX4) were markedly up-regulated, and markers of

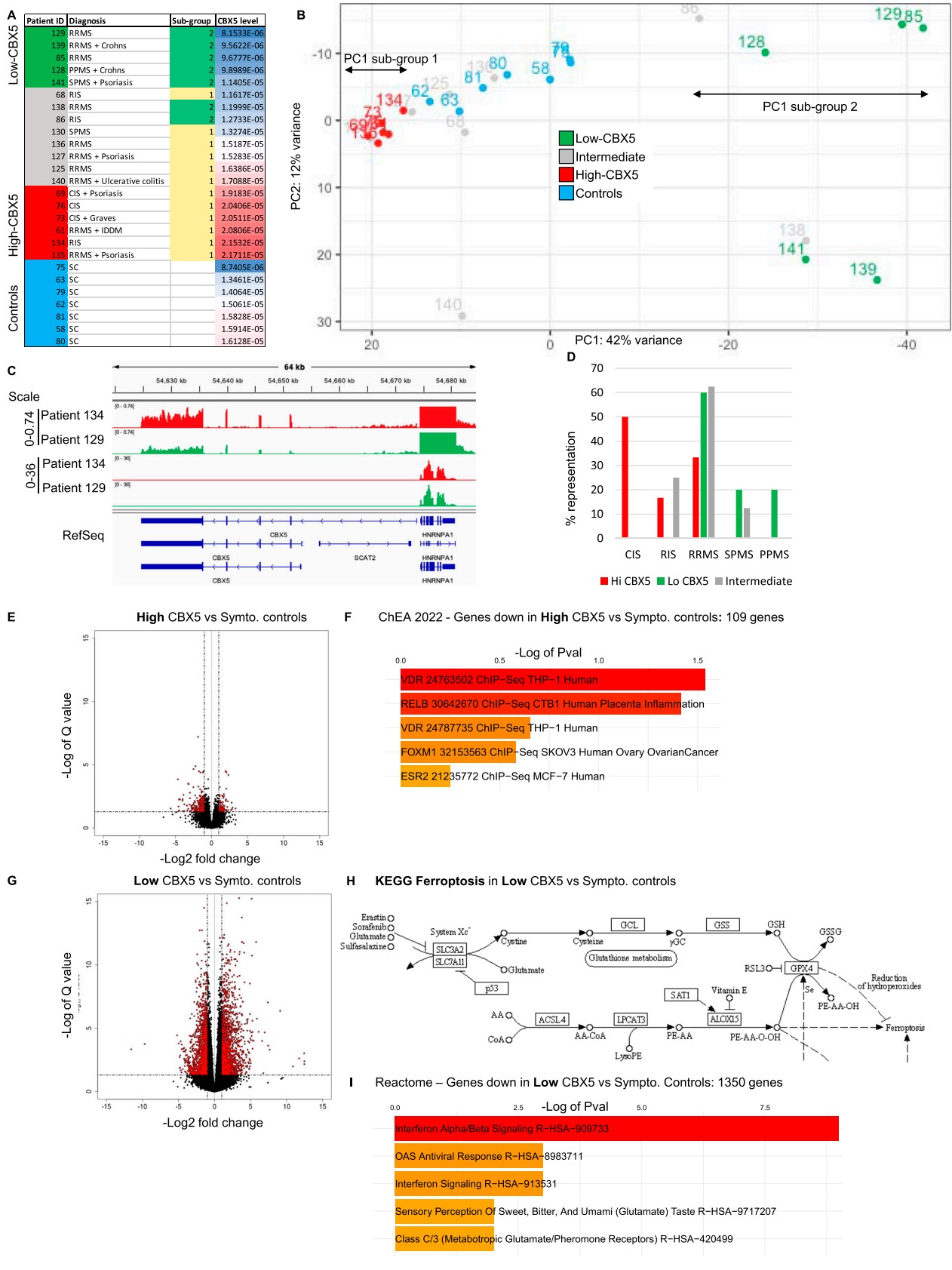

ferroptosis were either down-regulated (PTGS2) or unchanged (CHAC1) (Figs 1H and S1B). Although transcriptional activity is only indicative on enzymatic activities, this suggests that proferroptotic signals are activated in the Lo-CBX5 monocytes, but are offset by compensatory mechanisms inhibiting cell death.

Analysis of the down-regulated genes with either Reactome, BioPlanet, or WikiPathways databases revealed significant enrichment of genes associated with type I interferon signaling (Figs 1I and S1C). This is consistent with the observed benefits of administering interferon-alpha/beta to patients with MS (21).

In conclusion, most of the traits commonly linked with individuals with MS, such as heightened activation of stress pathways and diminished activity of type I interferon, are predominantly observed in patients who exhibit lower levels of CBX5 expression, in association with a significantly disrupted transcriptome. In contrast, patients expressing high levels of CBX5, enriched in patients in early phases of the disease, display only a minor deregulation of their transcriptome, exhibiting reduced activity of vitamin D reactive genes as their main characteristic.

## Evidence for reduced Integrator activity in Lo-CBX5 patients

To investigate the regulatory mechanism underlying the extensive transcriptome reprogramming observed in Lo-CBX5 patients, we confronted the list of genes up-regulated in Lo-CBX5 patients (relative to the SCs) with the ENCODE Transcription Factor Data Bank, which serves as a comprehensive repository of information about transcription factors and their involvement in controlling gene expression. This approach clearly designated NELFE targets as enriched among the up-regulated genes (Fig 2A). Prompted by this finding, we also examined transcriptome data from a large cohort of MS patient T cells (N = 118) available publicly (22). Segregating these patients into high and low CBX5 expressors revealed that Low-CBX5 T cells, similar to Lo-CBX5 monocytes, were primarily characterized by the altered expression of NELFE target genes (Fig S2A and D).

NELFE is a subunit of the negative elongation factor (NELF) complex, which controls the speed and efficiency of RNAPII movement along the DNA, and regulates gene expression by causing RNAPII pausing downstream of the transcription start site (TSS). As all RNAPII-transcribed genes require NELF, annotated NELF-targets designate genes at which promoter escape rather than transcription initiation is the rate-limiting step (23). The activity of NELF depends on the presence of at least two other complexes, the Integrator complex (INTcom) and the DRB sensitivity-inducing factor (DSIF) complex (24). In addition, the negative impact of these complexes on transcription is balanced out by factors having a positive effect on elongation, including the transcription factor MYC (25).

Examination of the transcriptome did not indicate any clear variations in the expression of the subunits of NELF. In contrast, 9 of 16 subunits of the INTcom were significantly (adj. P < 0.05) down-regulated (Fig 2B). Among these, INTS8 was previously reported to harbor SNPs associated with an elevated risk of MS (26, 27). Levels of INTS3 and INTS8 were also reduced in Low-CBX5 as compared to High-CBX5 T cells in the publicly available MS patient cohort (Fig S2B and C). The INTcom was initially characterized for its function in the 3′-end processing of U snRNAs, during which it recognizes the 3′ cleavage site and facilitates the cleavage reaction (28). In that context, we noted that an earlier study had reported aberrant U snRNA polyadenylation in patients with MS, a known manifestation of a defective maturation of these RNA species (29, 30, 31). Examination of our patient RNA-seq data with a genome browser revealed increased accumulation of 3′ extensions at U snRNA transcription at multiple loci, characteristic of a hampered maturation (Figs 2C and S2E and F). A compromised production of U snRNAs was also suggested by a defect in the maturation of several replication-dependent histone mRNAs, a process that relies on the U7 snRNA (32) (Fig 2D).

In parallel, the INTcom also promotes eRNA instability, recognizing and cleaving their 3′ end (33). We therefore quantified RNA-seq reads within annotated monocyte enhancers, revealing a significant increase in eRNA accumulation in Lo-CBX5, but not in Hi-CBX5 patients (Fig 2E). In this analysis, we also quantified RNA-seq reads mapping to DNA repeats, and neither SINEs, LINEs, nor endogenous retroviruses (HERVs) were up-regulated genome-wide in the patients (neither Lo-CBX5s, nor Hi-CBX5s) (Fig S2G–J). The unaltered transcription of DNA repeats also documents that the low levels of CBX5 did not result in an overall restructuring of chromatin-mediated transcriptional silencing. We noted, however, an up-regulation of HERVs coopted as enhancers, with transcription occasionally extending into DNA sequences encoding HERV-derived open reading frames (example in Fig 2F and quantification in Fig S2J). This suggested that the commonly observed up-regulation of HERVs in patients with MS might be attributed to a deficiency in terminating or processing eRNAs that originate from viral regulatory regions coopted as enhancers. A similar phenomenon also seemed responsible for the expression of protein-coding genes. For example, we noted that the leptin (LEP) gene, normally

---

**Figure 1. Extensive transcriptional deregulation in monocytes from patients with reduced expression of CBX5.**
**(A)** List of patients with MS with indications on the stage of the disease and eventual comorbidities. Symptomatic controls (SCs) are also indicated. Donors are ranked according to levels of CBX5 expression after normalization by DESeq2. Patients highlighted in red express CBX5 at high levels (upper quartile), whereas patients indicated in green express this gene at low levels (lower quartile). **(B)** PCA using the 500 most variable genes. **(C)** Graphic representation of the *CBX5-HNRNPA1* locus using the IGV genome browser. Bar charts represent the normalized expression levels of the two genes in the indicated patients. The two top lanes use a scale suited for the visualization of CBX5, whereas the two following lanes are at a scale fitted for HNRNPA1. RefSeq allows visualization of splice variants of the two genes annotated in the Hg19 version of the human genome. **(D)** Within each group of patients defined by either high, low, or intermediate CBX5 levels, the percentage representation of each stage of the MS disease is indicated. **(E)** Volcano plot illustrating differential gene expression between Hi-CBX5 patients and symptomatic controls. **(F)** Analysis of genes down-regulated between Hi-CBX5 patients as compared to symptomatic controls using the ChEA (ChIP-seq enrichment analysis) database providing information on transcription factors likely to regulate these genes (VDR, vitamin D receptor). **(G)** Volcano plot illustrating the differential gene expression between Lo-CBX5 patients and symptomatic controls. **(H)** Schematic representation of the KEGG ferroptosis pathway. Genes up- or down-regulated between symptomatic controls and Lo-CBX5 are indicated in red or blue, respectively. **(I)** Analysis of genes down-regulated in Lo-CBX5 compared with symptomatic controls using the Reactome pathway data bank revealing functional enrichment.

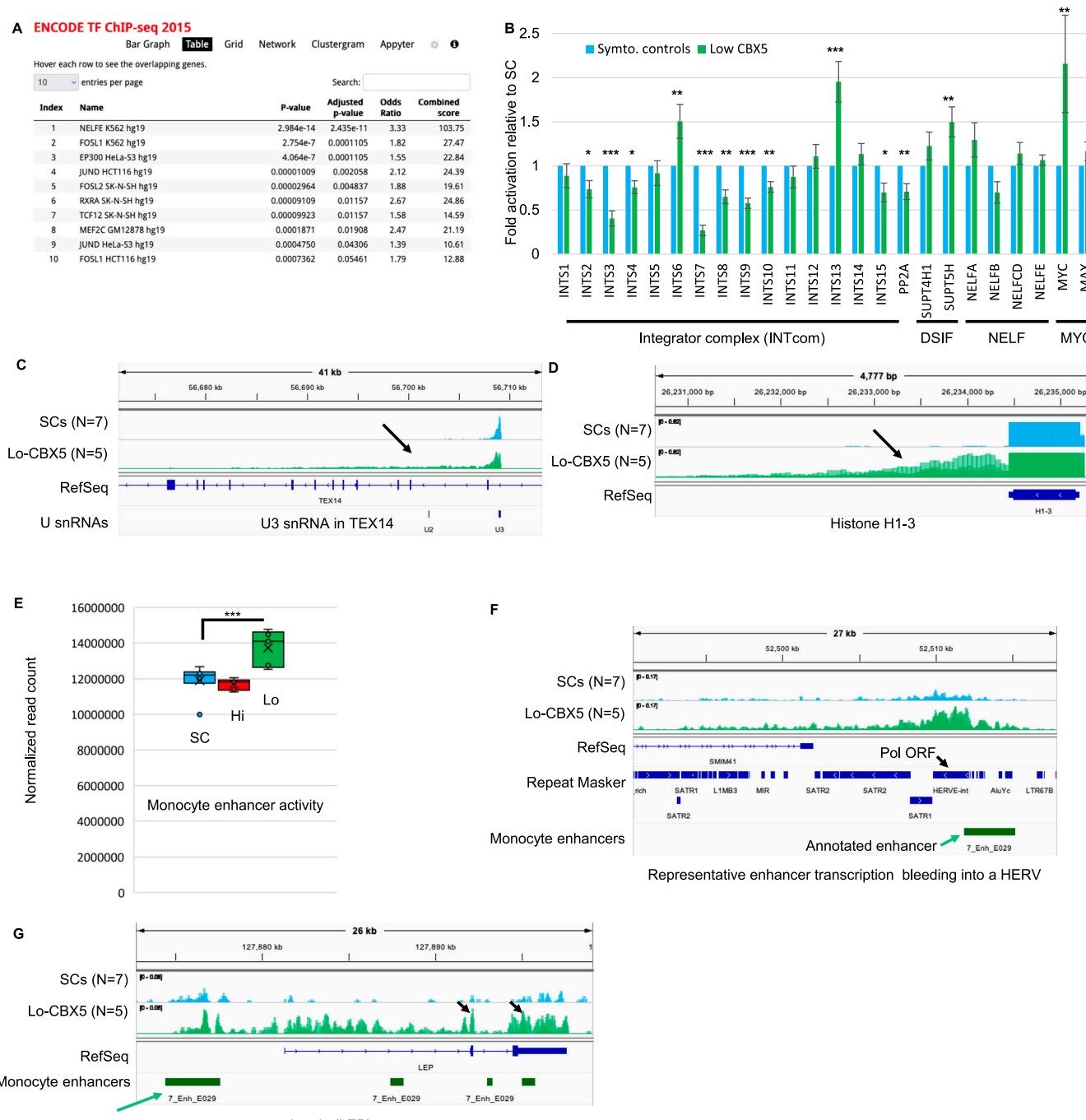

**Figure 2. Evidence for a defective Integrator activity in Lo-CBX5 patients.**
**(A)** Analysis of genes down-regulated between SC and Lo-CBX5 patients using the ENCODE TF ChIP-seq data resource providing insights into the binding patterns of various transcription factors. **(B)** Bar graph reporting variation in the expression of the indicated genes based on RNA-seq data from SCs and Lo-Cbx5 patients. Errors were calculated based on the lfcSE provided by DESeq2. Levels in SCs were set at 1. *Adj. *P* < 0.05, **adj. *P* < 0.01, ***adj. *P* < 0.001. **(C, D)** Screenshots from the IGV genome browser representing the expression of the U3 snRNA copy in the *TEX14* gene and of histone H1-3 (*HIST1H1D*) in SCs and Lo-CBX5 patients as indicated. **(E)** Quantification of reads mapping inside regions annotated as enhancers in monocytes by the Epigenome Roadmap. **(F, G)** Screenshots from the IGV genome browser representing the expression of enhancers producing eRNAs either covering several repeated sequences, including a HERV (F), or resulting in transcription of the entire *LEP* gene (G).

detected only in the adipose tissue and encoding a secreted hormone, was strongly activated in the Lo-CBX5 patient monocytes (sixfold increased expression, adj. *P* < 0.03), apparently as a consequence of eRNAs originating from an upstream enhancer elongating into the LEP gene body, and undergoing splicing (Fig 2G). Thus, the modified turnover of eRNAs appears as an unexpected source of ectopic gene expression in patients with MS.

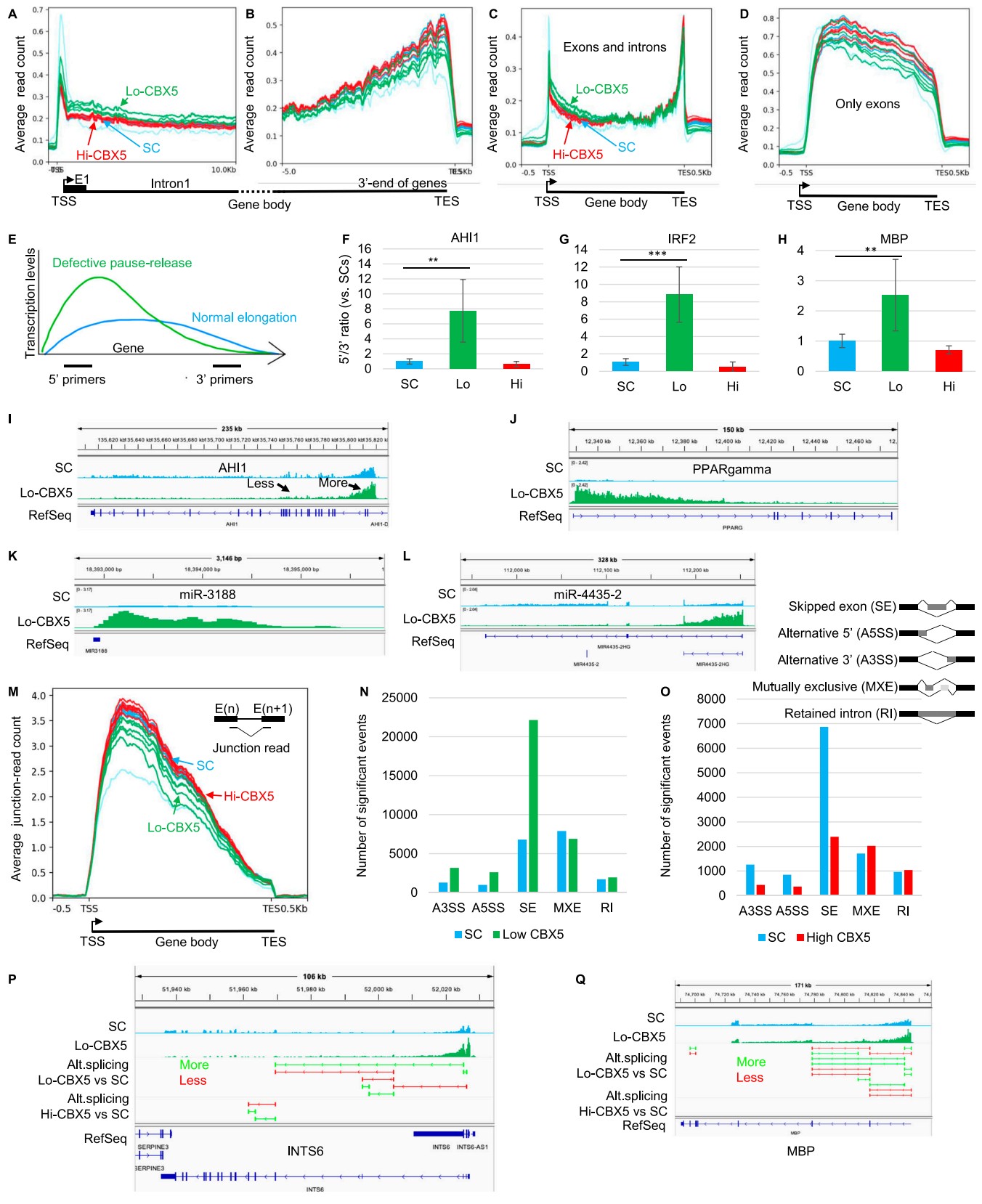

## Modified RNAPII pause–release affects the expression and splicing of genes relevant for MS

CBX5 activity is best documented in heterochromatin. However, CBX5 ChIP-seq data from HepG2 cells documented clear enrichment of this chromatin-bound protein at TSSs, suggesting that the chromatin regulator may also play a role in transcription initiation of protein-coding genes (Fig S3A). As mentioned above, the INTcom is an important player in the regulation of RNAPII pause–release, and reduced activity of the INTcom results in RNAPII entering the gene with reduced competence for elongation. This translates into exacerbated transcription at 5′ regions of genes, a phenomenon best observed with sequencing approaches detecting nascent pre-mRNA (34). However, visualization is also possible with RNA-seq data (35). To favor visualization of nascent RNAs in our RNA-seq patient data, we selected a set of genes with first introns exceeding 20 kb in length, allowing the monitoring of reads originating from pre-mRNA over a long, uninterrupted region. Average transcription profiles over these genes revealed consistently increased accumulation of reads downstream of the TSS in the Lo-CBX5 patients as compared to SCs and Hi-CBX5 patients, thereby mirroring the pattern observed in earlier in vitro studies (Fig 3A and (34)). This accumulation was not seen at the 3′ end of genes (Fig 3B).

As an alternative method to visualize the phenomenon, we generated metagenes with and without intronic sequences. When intronic sequences are excluded, the profiles show almost exclusively mature mRNA, more stable and therefore more abundant than pre-mRNA. Inversely, when including all sequences, the mature mRNAs account for only a minor fraction of the signal as exons are on average 300 nucleotides in length, whereas introns are several kilobases. These profiles confirmed increased transcription in the 5′ half of the genes in the Lo-CBX5 patients as compared to SCs and Hi-CBX5 patients (Fig 3C). In contrast, the mRNA output of the genes was on average equivalent in all donors (Fig 3D).

Finally, we used an RT–PCR approach to estimate the 5′/3′ transcription ratio at *AHI1*, *IRF2*, and *MBP* genes. This ratio is indicative of the competence of RNAPII for elongation, based on the principle that RNAPII poorly competent for elongation will transcribe the 5′ end of genes more than their 3′ end (schematic in Fig 3E). Because the monocyte RNA preparations were fully used for the RNA-seq approach, the PCRs were performed on RNA from the leftover PBMCs, enriched in B and T cells. Similar to the monocytes,

the remaining PBMCs from Lo-CBX5 MS patients displayed the reduced expression of CBX5, INTS3, and INTS8 compared with the remaining PBMCs from SC and Hi-CBX5 patients (Fig S3B). Comparing PCR products at both ends of the genes under scrutiny revealed a clear increase in the proportion of 5′ over 3′ transcription in Lo-CBX5 patients, in contrast to Hi-CBX5 patients and the SC group (Figs 3F–H and S3C–E). These experiments also further support that reduced Integrator activity and elongation defects are not restricted to monocytes.

We next explored how defective RNAPII elongation affected the transcriptome of the patients. First, we noted that genes activated in Lo-CBX5 patients were on average shorter than the downregulated genes, an expected manifestation of globally increased transcription of regions located immediately downstream of the TSS (Fig S3F). Next, examination of the RNA-seq data with a genome browser allowed identifying multiple genes relevant for MS and displaying increased transcription over the first exon and the first intron, with consequences on their global level of expression. For example, the *AHI1* gene, for which reduced expression has previously been associated with an increased risk of MS (36), showed a clear increase in transcription downstream of the TSS ("More" arrow, Fig 3I), whereas transcription was reduced from exon 5 onward ("Less" arrow, Fig 3I). This suggested that reduced control of RNAPII pause–release at this gene resulted in decreased production of full-length pre-mRNA. At other genes, uncontrolled promoter escape resulted in a net increase in the production of mature mRNA. This was exemplified by the *PPARG* and *SLC7A11* genes, mostly transcribed in their 5′ regions, yet overall activated respectively sixfold and threefold according to DESeq2 quantification (adj. $P < 0.05$; see representative patients in Figs 3J and S3G). The strong initiation/poor elongation behavior of the RNAPII also modified the expression of regulatory RNAs when embedded in larger genes. For example, it caused the increased expression of miR-3188 previously described as an MS risk factor (37, 38) (Fig 3K), while resulting in the decreased expression of miR-4435-2, a miR located further inside its host gene (Fig 3L).

From earlier studies, MS is known also to affect alternative splicing (AS) at many genes (39). Therefore, we examined whether the MS patients exhibited any alterations in overall splicing levels, using the RNA-seq data to quantify junction reads (reads produced when the sequenced RNA fragments span over exon–intron boundaries—such reads serve as evidence of a splicing event). This approach showed that on average, less splicing events were

**Figure 3. Defective Integrator activity results in extensive gene deregulation.**
**(A, B, C, D)** Average distribution of RNA-seq reads at genes containing an initial intron exceeding 20 kilobases in length. **(C, D)** Profiles are either anchored on the transcription start site (TSS in A) or on the transcription end site (TES in B), or plotted over the entire metagene either including (C) or excluding (D) intronic sequences. SC, symptomatic control; Hi-CBX5, patients in the upper quartile for CBX5 expression levels; Lo-CBX5, patients in the lower quartile for CBX5 expression levels. **(E)** Schematic representation of the position of the primers used to evaluate transcription in 5′ and -3′ regions of the genes under scrutiny. Curves represent the expected distribution of transcription during normal elongation (blue) and upon accumulation of the RNAPII in promoter-proximal regions because of defective pause–release (green). **(F, G, H)** For each SC (N = 6), or Lo-CBX5 (N = 4) or Hi-CBX5 (N = 6) MS patient, RNAs were purified from PBMCs remaining after monocyte isolation and used for RT–qPCR with primer pairs targeting either the 5′ or 3′ regions of the indicated genes. The RT–qPCR data are presented in Fig S3C–E. For each patient, these PCRs were then used to calculate a 5′/3′ ratio. The bar graphs show the median values of these ratios. **(I, J, K, L)** Screenshots from the IGV genome browser representing the impact of deregulated RNAPII pause–release at the indicated genes. **(C, M)** Average distribution of junction reads from the RNA-seq data across a metagene as in (C). **(N, O)** RNA-seq was analyzed with rMATS to quantify alternative splicing occurring between SC and Lo-CBX5 patients (N), or between SC and Hi-CBX5 patients (O). Histograms indicate the number of significant events ($P < 0.05$). A3SS, alternative 3′ splice site; A5SS, alternative 5′ splice site; SE, skipped exon; MXE, mutually exclusive exons; RI, retained introns. **(P, Q)** Screenshots from the IGV genome browser representing, at the indicated genes, splicing events either up-regulated (green) or down-regulated (red) in the Lo-CBX5 (top track) or in the Hi-CBX5 (bottom track) when compared to SCs ($P < 0.05$).

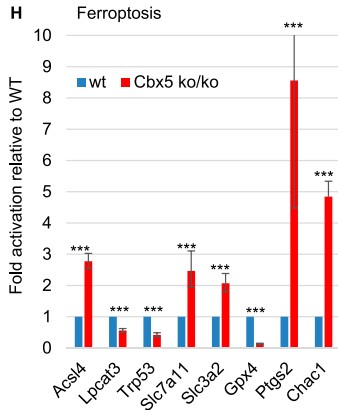

**A**

MOG$_{35-55}$/CFA s.c.
+ Pertussis toxin i.p.

Day 0 → Day 2

**B**

Mean clinical score (y-axis, 0–5)
Days after immunization (x-axis, 1–31)

—◆— WT    —■— Cbx5 KO    —▲— Control

**C**  Up-genes

Cbx5 KO: 2527
703 (common)
EAE: 720

**D**  703 common up genes

-Log10(p-val) (0–6)

Cytokine–cytokine receptor interaction
NF–kappa B signaling pathway
IL–17 signaling pathway
Amoebiasis
Ribosome biogenesis in eukaryotes

KEGG 2021

**E**

-Log10(p-val) (0.0–7.5)

MAX myocyte mm9
MYC MEL cell line mm9
MAX MEL cell line mm9
MAX C2C12 mm9
MAX endothelial cell of umbilical vein hg19

ENCODE TF ChIP-seq 2015

**F**  3230 genes up in Cbx5-/- vs WT (Min. 2-fold, adj. pVal<0.05)

-Log10(p-val) (0–2)

Transport Across Blood–Brain Barrier (GO:0150104)
Vascular Transport (GO:0010232)
Ribonucleoprotein Complex Biogenesis (GO:0022613)
Ribosome Biogenesis (GO:0042254)
L–alpha–amino Acid Transmembrane Transport (GO:1902475)

GO Biological Process 2023

**G**  77 SLC genes upregulated (Min. 2-fold, Adj pVal<0.05)

-Log10(p-val) (0–40)

L–amino Acid Transmembrane Transporter Activity (GO:0015179)
Carboxylic Acid Transmembrane Transporter Activity (GO:0046943)
Amino Acid Transmembrane Transporter Activity (GO:0015171)
Neutral L–amino Acid Transmembrane Transporter Activity (GO:0015175)
Antiporter Activity (GO:0015297)

GO Molecular Function 2023

**H**  Ferroptosis

Fold activation relative to WT (y-axis, 0–10)

■ wt    ■ Cbx5 ko/ko

Acsl4 ***, Lpcat3 ***, Trp53 ***, Slc7a11 ***, Slc3a2 ***, Gpx4 ***, Ptgs2 ***, Chac1 ***

detected in Lo-CBX5 patients when compared to Hi-CBX5 and SC donors, with a more pronounced impact at the 5′ end of genes (profiles in Fig 3M, green below blue and red). Further analysis of the RNA-seq data with the rMATS package confirmed that alternative splicing was affected in the MS patients, whether displaying Lo- or Hi-CBX5 expression. Yet, it also documented that the reduced splicing level in the Lo-CBX5 patients is essentially translated into exon skipping, consistent with the fact that fewer splicing events are needed when fewer exons are included (Fig 3N). Inversely, Hi-Cbx5 patients displayed reduced exon skipping when compared to the SCs (Fig 3O). Interestingly, increased exon skipping in Lo-CBX5 patients was observed at the INTS6 gene, encoding a regulatory subunit of the INTcom (40). This suggested that reduced Integrator activity is a self-reinforcing phenomenon (Fig 3P). Increased exon skipping was also observed at MBP and GAPDH genes encoding MS autoantigens (Figs 3Q and S3H).

Thus, reduced Integrator activity translates into either up- or down-regulation of numerous genes associated with multiple sclerosis (MS), the outcome possibly varying depending on the extent to which these genes rely on RNAPII pause–release for their regulation. In addition, gene expression is further impacted by overall reduced splicing.

## Inactivation of Cbx5 in the mouse promotes inflammation and exacerbates experimental autoimmune encephalomyelitis (EAE)

To explore further a possible causative role of reduced CBX5 activity in the MS pathogenesis, we implemented a mouse model inactivated for the Cbx5 gene. This mouse model was viable, although animals homozygous for the Cbx5 mutation (Cbx5-/-) displayed high perinatal mortality.

We first challenged the mouse model with EAE, a protocol widely used to mimic some aspects of MS by triggering an autoimmune response with injections of myelin-derived peptides followed by induction of inflammation by administration of pertussis toxin (Fig 4A). Cbx5−/− mice displayed an exacerbated reaction to that protocol, with an earlier onset of the symptoms followed by a rapid reaching of high-grade EAE, beyond recovery (Fig 4B). The early onset and the exacerbated symptoms were also observed with heterozygous Cbx5+/− mice, and these animals experienced a more extensive loss of body weight than the wild types during the protocol (Fig S4A and B). Unlike the Cbx5−/−, the Cbx5+/− mice eventually transitioned into a recovery phase, although with a 1-d delay compared with the wild types (Fig S4A).

We next carried out RNA-seq on CD4$^+$ T cells from either WT (n = 2), Cbx5+/− (n = 1), Cbx5−/− (n = 3), EAE (n = 3), or Cbx5−/− EAE (n = 3).

As anticipated from the Lo-CBX5 MS patients, inactivation of Cbx5 had an extensive impact on the transcriptome, with a total of 6,478 differentially regulated genes (3,224 up- and 3,248 down-regulated genes twofold or more, adj. $P$ < 0.05—Fig S4C and D and Table S2A–C). Among the 3,224 up-regulated genes, 703 genes were also up-regulated by EAE, corresponding to approximately half of the genes up-regulated by EAE (Fig 4C). This observed intersection was approximately sixfold higher than expected by chance (Fig S4E). Pathway analysis on the 703 shared genes revealed that Cbx5-/- mice activated IL-17 and NFkappaB pathways at the steady state, which may participate in the exacerbated response to the EAE protocol (Fig 4D). We noted also that the 703 genes up-regulated both by Cbx5 inactivation and by EAE were enriched in Myc target genes (Fig 4E). This observation suggested a potential reliance of these genes on regulation through RNAPII pause–release mechanisms (41).

Gene ontology analysis of the 3,224 genes up-regulated after Cbx5 inactivation identified strong enrichment scores for transmembrane transport through the blood–brain barrier, underscoring the modified expression of numerous solute carrier (SLC) genes (Fig 4F). These genes encode membrane transport proteins involved in the movement of ions, nutrients, and metabolites and frequently associated with inflammation and autoimmunity disease (42). We noted that among the 373 SLC genes annotated in the mouse genome, 117 displayed a modified expression (min. twofold, adj. $P$ < 0.05), whereas the 77 up-regulated SLC genes were enriched in amino acid transporters (Fig 4G). These observations suggest that SLC genes are particularly sensitive to Cbx5 regulation and that the modified expression of the SLC genes may promote T-cell activation, heavily relying on amino acid transportation (43).

To mirror our study on the MS patients, we also examined the ferroptosis pathway. We here noted a strong down-regulation of Gpx4 (10-fold) and a corresponding up-regulation of the ferroptosis markers Ptgs2 and Chac1 (Fig 4H). Consistent with a possibly increased cell death by ferroptosis, we noted also that down-regulated genes (twofold or more, adj. $P$ < 0.05, 3,248 genes) were enriched in genes involved in fatty acid beta-oxidation (FAO—Fig S4F). A down-regulation of this pathway is anticipated to favor ferroptosis, as FAO normally consumes the fatty acids, leading to a reduction in the rate of lipid peroxidation (44). Analysis of the down-regulated genes with either Reactome or BioPlanet databases revealed that these genes were also enriched in genes involved in cell cycle regulation, DNA replication, and mitochondrial ATP synthesis, consistent with a subset of the cells possibly withdrawing from the cell cycle (Fig S4G and H).

**Figure 4. Inactivation of Cbx5 in the mouse exacerbates EAE.**
**(A)** Schematic representation of the EAE protocol. **(B)** Graphic representation of the clinical scores observed for the mice included in the EAE protocol: 0, no clinical symptoms; 1, flaccid tail (loss of tail tone); 2, hindlimb weakness or partial paralysis; 3, complete hindlimb paralysis; 4, complete hindlimb paralysis and partial forelimb paralysis; and 5, moribund or dead. Indicated data are averaged from 10 WT mice and 5 Cbx5 −/− mice, 10 wk of age. Error bars represent the SEM. **(C)** Venn diagram representing the number of genes up-regulated either in Cbx5 null mice compared with WT, in mice exposed to an EAE protocol compared with controls, or in both conditions. **(D, E)** Analysis of genes up-regulated both upon Cbx5 inactivation and by the EAE protocol using the KEGG database providing information about the molecular functions and biological pathways of genes (D), and the ENCODE TF ChIP-seq data resource providing insights into the binding patterns of various transcription factors (E). **(F)** Analysis of genes up-regulated upon Cbx5 inactivation using the "Biological Process" category in the GO database focusing on describing biological events, pathways, and processes. **(G)** Analysis of the 77 SLC genes up-regulated (Min. twofold, adj. $P$ < 0.05) upon Cbx5 inactivation using the "Molecular Function" category in the GO database focusing on the molecular activities of gene products. **(H)** Bar graph reporting the variation in the expression of the indicated genes based on RNA-seq data from WT and Cbx5−/− mice. Levels in WTs were set at 1. *Adj. $P$ < 0.05, ** adj. $P$ < 0.01, ***adj. $P$ < 0.001.

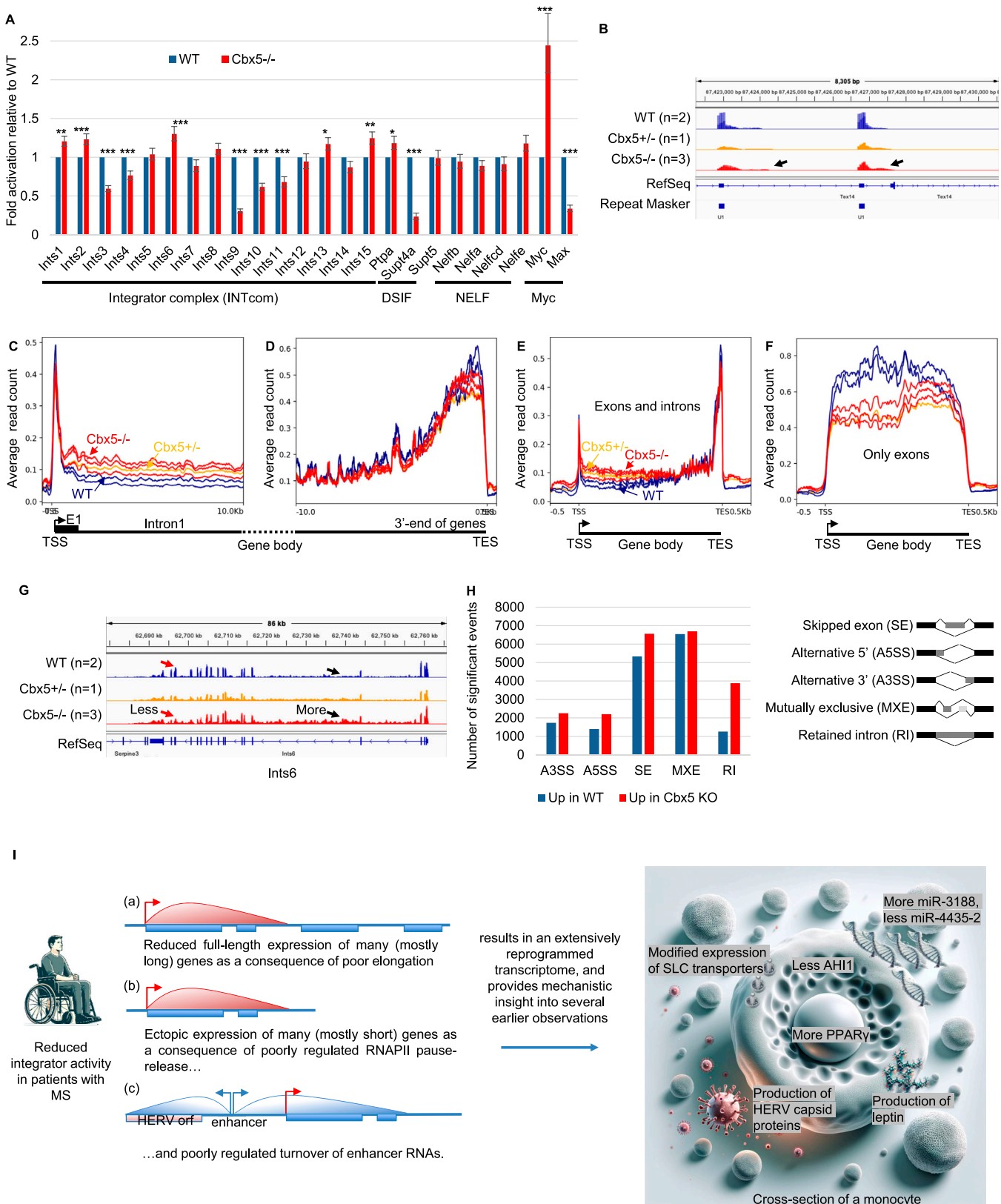

**Figure 5. Mouse model confirms the impact of Cbx5 on Integrator activity.**
**(A)** Bar graph reporting the variation in the expression of the indicated genes based on RNA-seq data from WT and Cbx5−/− mice. Errors were calculated based on the lfcSE provided by DESeq2. **Adj. *P* < 0.01, ***adj. *P* < 0.001. **(B)** Screenshots from the IGV genome browser representing the expression of two U1 snRNA copies in WT,

## The *Cbx5−/−* mouse model recapitulates the defect in Integrator activity

Finally, we investigated whether *Cbx5−/−* mice recapitulated some of the transcriptional anomalies observed in Lo-CBX5 MS patients. Examination of the transcriptome revealed the reduced expression of several INTcom subunits (Ints3, Ints4, Ints9, Ints10, and Ints11/Cpsf3l), and of the DSIF subunit Supt4h1/Spt4, whereas the expression of Myc was increased (Fig 5A). We noted also that on average, U snRNA genes showed reduced expression, with some loci exhibiting disproportionately high downstream transcription, consistent with poor maturation of these RNA species (see examples in Fig 5B and average profile in Fig S5A). Similarly, histone genes, which also rely on the Integrator complex for their maturation, were abundantly down-regulated. Specifically, 62 of 70 annotated histone genes were significantly down-regulated ($P < 0.05$), and as observed for the snRNAs, there was disproportionately high transcription in the downstream regions at some loci (Fig S5B and C). We note, however, that the reduced expression of histone genes may also be linked to the previously mentioned reduction in cell cycling.

To probe for defects in the regulation of RNAPII pause–release, we plotted the average distribution of reads at a set of genes harboring a first intron of more than 20 kb, thereby using the same approach as for the MS patient samples. For the 3 *Cbx5−/−* samples and the one *Cbx5+/−*, alike what we observed in the Lo-CBX5 patients, accumulation of reads was increased over the region downstream of the TSS, while unaffected or reduced over the 3′ region of the genes (Fig 5C and D). The increased transcription at the TSS-proximal regions was also visualized when plotting the distribution over an average (meta)gene including both exons and introns (Fig 5E). In contrast, plotting the read distribution over exons only revealed an overall reduced production of mature mRNA, consistent with the mouse model recapitulating a reduced efficiency of splicing (Fig 5F). The Ints6 gene provided a clear illustration of this transcriptional defect, with accumulation of intronic reads, associated with a reduced signal at exons, particularly toward the 3′ end of the gene (Fig 5G). Analysis of the data with the rMATS package confirmed an extensive impact of *Cbx5* inactivation on alternative splicing, with, as observed in the Lo-CBX5 patients, an increase in exon-skipping events. Yet, consistent with an accumulation of non-maturated pre-mRNA species, the most favored type of alternative splicing was intron retention (Fig 5H). EAE alone reduced rather than increased accumulation of reads on the 5′ region of genes (Fig S5D). Yet, like in the untreated animal, *Cbx5* inactivation resulted in accumulation of reads downstream of TSSs (Fig S5E).

# Discussion

Our RNA-seq analysis of monocytes from a relatively diverse patient cohort revealed that CBX5 expression levels, rather than comorbidity, age, or gender, partitioned patients into two distinct groups. Patients with low CBX5 expression showed significant transcriptional differences from control donors, whereas those with high CBX5 expression resembled the controls more closely, with a transcriptome mainly characterized by the reduced expression of vitamin D–regulated genes. The correlation between the level of serum vitamin D and disease activity still needs to be further investigated (45). Yet, this observation suggests that the Hi-CBX5 group, which included individuals with clinically and radiologically isolated syndromes, might particularly benefit from a vitamin D–enriched diet.

The Lo-CBX5 group was enriched in patients in primary and secondary progressive stages, while devoid of isolated syndromes, suggesting that low CBX5 expression may be indicative of a steady neurological decline. However, we note that our sample size was too small to draw definitive conclusions about the characteristics indicated by Lo-CBX5 expression. Specifically, patients in progressive stages are also more likely to be in an active phase of the disease at the time of sampling, a factor that may also be highly relevant. At the transcriptional level, the Lo-CBX5 group was characterized by defects in U snRNA and eRNA processing and in RNA polymerase II (RNAPII) pause–release, collectively indicative of reduced Integrator complex activity. A link between the reduced CBX5 expression and the altered expression of genes where RNAPII pause–release is a rate-limiting step was also observed in MS patient T cells, as demonstrated by reanalyzing a set of published RNA-seq data (22). An implication of the Integrator complex in the MS pathology has hitherto not been investigated, although several indications have been pointing in that direction. Firstly, SNPs within the Integrator subunit INTS8 have previously been associated with an elevated risk of MS (26, 27). Another study has also detected an association between MS and SNPs in genes associated with "abortive elongation," including the NELF subunits NELFE and NELFCD, and the DSIF subunit SUPT4H1 (46). Finally, patients with MS have been reported to show aberrant U snRNA polyadenylation, a known manifestation of a defective maturation of these RNA species (29, 30, 31), whereas we have previously described

---

*Cbx5+/−*, and *Cbx5−/−* mice as indicated. Arrows indicate the presence of non-maturated transcripts. **(C, D, E, F)** Average distribution of RNA-seq reads at genes containing an initial intron exceeding 20 kilobases in length. **(C, D, E, F)** Profiles are either anchored on the transcription start site (TSS in (C)) or on the transcription end site (TES in (D)), or plotted over the entire metagene either including (E) or excluding (F) intronic sequences. **(G)** Screenshots from the IGV genome browser representing the impact of deregulated RNAPII pause–release at the *Ints6* gene. **(H)** Histograms indicate the number of significant events ($P < 0.05$). A3SS, alternative 3′ splice site; A5SS, alternative 5′ splice site; SE, skipped exon; MXE, mutually exclusive exons; RI, retained introns. **(I)** Model: a subset of patients with MS, particularly those in progressive phases of the disease, exhibit reduced Integrator activity. This leads to a loss of control over RNAPII pause–release, resulting in increased transcription at promoter-proximal regions and decreased transcription in more distal regions. Consequently, this causes the reduced expression of some genes (a) and activation of others (b), with the impact depending not only on the size of the gene but also on the reliance of each gene on regulation at the level of RNAPII pause–release. In parallel, the reduced Integrator activity also affects the control of eRNA maturation (c), and transcripts initiated at enhancers occasionally extend into neighboring genes, eventually of viral origin (GAG, Pol, Env). These transcriptional anomalies may explain numerous transcriptional events previously associated with MS, including the production of retrovirus-like particles, altered leptin production, changes in the expression of various miRNAs, the altered expression of SLC11A7/xCT and AHI1, and the increased expression of PPARγ.

increased accumulation of eRNAs associated with MS (11). More generally, we note that the Integrator complex is a known player in neurological diseases, with mutations in INTS1 and INTS8 being associated with rare recessive human neurodevelopmental syndromes (47).

Reduced Integrator activity affects the expression of hundreds of protein-coding genes independently of its impact on snRNA maturation (34, 48). One mechanism enabling this effect involves restricting pause–release to elongation-competent RNAPII enzymes. The Integrator complex therefore promotes the production of full-length pre-mRNAs. Conversely, limited Integrator activity biases transcription toward promoter-proximal regions (illustrated in the model in Fig 5I). A distribution of reads indicative of diminished Integrator activity was observed at the *AHI1* gene, peaking at the initial exons, then rapidly decreasing (Fig 3I). This suggests that the reduced expression of AHI1, which is strongly linked to a higher risk of MS, might be due either to SNPs as previously reported (36) or to decreased Integrator activity. In parallel, poorly controlled RNAPII pausing up-regulates basal transcription of (often short) inducible genes (49). Therefore, the chronic activation of short intronless proinflammatory genes such as c-Jun and JunD may be directly related to this mechanism, consistent with up-regulated genes being on average shorter than down-regulated genes in the Lo-CBX5 patients. Longer genes affected by this mechanism include *PPARG* and *SLC7A11* (xCT) and may be activated in the context of a protective effect, PPARG modulating the immune response (50), while SLC7A11 promoting cell death as will be discussed below (51). Finally, the RNAPII incompetent for elongation also accounts for the observed down-regulation of miR-3188 and up-regulation of miR-4435-2, respectively located either at the start of the host gene, receiving excessive RNAPII, or in a more central region, thus being out of reach of the polymerase (37, 38).

Deregulated elongation of eRNAs, another hallmark of reduced Integrator activity, emerges as a second mechanism promoting ectopic gene expression (refer to the model in Fig 5I). A prime example of this is the increased transcription of HERV-encoded sequences. Promoter regions of ancient viruses inserted in the human genome are occasionally coopted as enhancers (52). We observe that at some of these enhancers of viral origin, the increased elongation of eRNAs mechanically results in the transcription of the adjacent viral genes. This phenomenon may possibly explain the increased expression of HERV envelope proteins at the surface of monocytes and the production of retrovirus-like particles from cultured patient cells (53, 54). The impact of elongated eRNAs was also exemplified by the leptin gene that we found transcribed from an upstream enhancer elongating into the *LEP* gene. This adipokine normally secreted by adipose tissues regulates systemic metabolism and appetite, while also signaling directly to immune cells to promote inflammation (55). Leptin has been found to be present at increased levels in patients with MS and has been associated with MS risk (56, 57). This may directly illustrate how defective Integrator activity can participate in the pathogenesis of MS.

Reduced Integrator activity was the main transcriptional phenotype of the mouse T cells inactivated for Cbx5, suggesting that the encoded HP1α protein is directly involved in the regulation of RNAPII activity. This is in contrast to previously documented heterochromatic activity of this protein. Yet, examination of the available ChIP-seq data revealed a localization of HP1α at TSSs of genes. We note also that HP1α is known to dimerize with TRIM28/KAP1 previously reported to participate in the regulation of RNAPII elongation (58, 59). Finally, HP1 proteins copurifies with the CBC cap-binding complex, consistent with a role in early phases of transcription initiation (60). In an earlier study, we had shown that HP1α activity was affected by increased citrullination of histone H3R8 (10). In the monocytes of the cohort of MS patients examined here, we did not observe any increased transcription of PADI enzymes responsible for this citrullination event. This suggests that HP1α activity may be compromised via either transcriptional or post-transcriptional mechanisms, possibly as a function of the cell type under scrutiny.

The mechanism causing *Cbx5* inactivation to promote EAE still needs to be explored at an immunological level to identify the involved cell populations. Yet, we note that the many genes up-regulated by both EAE and *Cbx5* inactivation were enriched in Myc targets. The Myc transcription factor is indispensable for T-cell activation, playing a role in the amino acid supply required for protein production associated with the increased cellular activity (61). The amino acid transporter Slc7a5, a key target of Myc, is also among the numerous SLC genes up-regulated by *Cbx5* inactivation (>fourfold, adj. $P = 10^{-80}$). This apparently exacerbated sensitivity of genes of the SLC family to deregulated RNAPII pause–release may possibly explain some aspects of the increased responsiveness of Cbx5 null mice to EAE.

While displaying numerous similarities, the transcriptional defects observed in Lo-CBX5 patients and in Cbx5 null mice were not entirely identical. In particular, in the mouse model, we noted a reduced expression of U snRNAs and a more pronounced disruption of pre-mRNA maturation. It is plausible that these two phenomena are related. The significant scarcity of U snRNAs in the mouse model could be a major factor contributing to the extensive intron retention. In contrast, in patients, where the CBX5 deficit is milder, relatively less affected splicing machinery may cause only increased exon skipping.

Whether reduced CBX5 activity is a trigger of MS symptoms remains an open question. We noted that in the Cbx5 KO mouse model, inactivation of just one *Cbx5* allele was sufficient to obtain both the reduced Integrator activity and the exacerbated reaction to EAE. Nevertheless, the mice did not manifest EAE symptoms spontaneously, suggesting that *Cbx5* inactivation generates a conducive terrain, but that the neuroinflammatory symptoms necessitate an external trigger to develop. Although our data do not offer insights into the potential nature of this trigger, they may provide valuable information regarding subsequent transcriptional events. Indeed, several observations suggest that reduced Integrator activity is a self-amplifying mechanism. First, mutations in INTS1 and INTS8 were previously shown to result in the down-regulation of multiple subunits of the INTcom (47). A similar phenomenon was observed upon *Cbx5* inactivation in our mouse model, with down-regulation of Ints3, Ints4, Ints9, Ints10, and Cpsf3l/Ints11. The reduced expression of these INTcom genes may be due to a sensitivity of these genes to their own activity, as we documented for the *INTS6* gene in both human and mouse. This suggests that external stimuli hitting on RNAPII pause–release may

trigger a feedforward loop resulting in reduced INTcom activity. In this context, we note that the EBV was found associated with multiple sclerosis (MS), first described 30 yr ago (62) and has continued to show evidence as a causal role in MS (63). This virus is known to target INTS6 (64), while also being a promoter of ferroptosis (65). Together with the clear evidence for ferroptosis in the *Cbx5−/−* mice, this may suggest that ferroptosis is the normal fate of lymphocytes experiencing a defective Integrator activity. The seemingly abortive ferroptosis in the MS patients may further suggest that this defense mechanism is not entirely functional in these patients, possibly participating in the exacerbated neuroinflammation.

# Materials and Methods

### Ethics statement

The study was conducted in accordance with the Ethical Declaration of Helsinki, and all patients gave written, informed consent. The study and the material for informed consent were approved by the Central Denmark Region Committee on Biomedical Research Ethics (journal number: 1-10-72-334-15).

### Patients and controls

Patients admitted to the MS Clinic, Department of Neurology, Aarhus University Hospital, were consecutively included from January 2017 to January 2018. A full diagnostic workup included medical history, clinical examination, MRI of the entire neural axis, cerebrospinal fluid (CSF) analyses (cells, protein, IgG index, oligoclonal bands), and evoked potentials as recommended (66). CSF and MRI examinations were evaluated according to the revised MacDonald criteria from 2017 (66), and an Expanded Disability Status Scale score was assessed according to Kurtzke (67). Patients were excluded if they had other neurological diseases or received glucocorticoids within the month preceding sampling. A total number of MRI white matter lesions were registered by fluid-attenuated inversion recovery sequences on MRI. Demographics and paraclinical findings of patients with clinically isolated syndrome, RRMS, PPMS, and RIS patients, and SCs are summarized in Table S1D. Patients included as SC have neurological symptoms, but have no objective clinical or paraclinical findings to define a specific neurological disease. This specific definition is described in detail by reference 68, and they do not represent early MS.

Patients included as RIS have no neurological symptoms and are only referred to diagnostic workup owing to the presence of incidental white matter lesions in MRI suggestive of MS. Diagnostic criteria for RIS were proposed in 2009 and include the number, shape, and location of the brain lesions (66). Lesions are ovoid and well circumscribed with a size greater than 3 mm, show dissemination in space, and can be juxtaposed to the corpus callosum. Lesions should not follow a vascular distribution and do not account for any other pathologic processes.

### Data download, pathway analysis, and data visualization

Analysis of MS patient CD4$^+$ T cells (22): NCBI-generated raw count matrix was downloaded here: https://www.ncbi.nlm.nih.gov/geo/download/?type=rnaseq_counts&acc=GSE137143&format=file&file=GSE137143_raw_counts_GRCh38.p13_NCBI.tsv.gz, and used to sort patients according to levels of CBX5 expression. We then used the GEO2R tool (https://www.ncbi.nlm.nih.gov/geo/geo2r/) to compare the upper quartile (highest CBX5 expression) and the lower quartile (lowest CBX5 expression). Genes differentially regulated (up or down) with an adjusted *P*-value < 0.05 were obtained from the included "volcano plot" tool. Details on the patients included in each quartile, on the genes differentially expressed between these quartiles, and on the output of the analysis of the processed ChIP-seq data from ENCODE using Enrichr are provided in Table S3. Cbx5 ChIP-seq data acquired in hepatocarcinoma-derived HepG2 cells were downloaded from the ENCODE portal; output type was "signal *P*-value" in bigWig format, file ENCFF408KOE (https://www.encodeproject.org/) (69). Enrichr, used for pathway analysis, is made available by the Ma'ayan Lab (70, 71). The Integrative Genomics Viewer (IGV) software was used to examine specific loci (72).

### RNA sequencing

Patient PBMCs were isolated as previously described (18). Monocytes were then isolated using Pan Monocyte Isolation Kit (ref. 130-096-537; Miltenyi Biotec) following the kit protocol, allowing for negative selection of unstimulated monocytes. Purity of the monocytes was verified by flow cytometry on patient #129, based on cell size and granularity. CD4$^+$ mouse T cells were purified from splenocytes using the Miltenyi kit 130-104-454. For both cell types, total RNA was extracted by TRIzol LS (ref. 10296028; Thermo Fisher Scientific), according to the manufacturer's protocol. Total RNA library preparation and sequencing were performed by Novogene Co., Ltd, as a lncRNA sequencing service, including lncRNA directional library preparation with rRNA depletion (Ribo-Zero Magnetic Kit), quantitation, pooling, and PE150 sequencing (30G raw data) on the Illumina HiSeq 2500 platform. Filtering and trimming of the RNA-seq data left around 230–300 million read pairs/sample. Mapping was carried out with STAR (v2.6.0b) (parameters: --outFilterMismatchNmax 1 --outSAMmultNmax 1 --outMultimapperOrder Random --outFilterMultimapNmax 30) (73). The reference genomes were hg19 Homo sapiens and mm10 Mus musculus primary assemblies from Ensembl. The SAM files were converted to BAM files and sorted by coordinate using SAMtools (v1.7) (74).

### bigWig files, heatmaps, and profiles

The bigWig files were generated from .bam files with bamCoverage (parameter: --normalizeUsing CPM) from deepTools (v3.1.3) (75). Heatmaps and profiles were also generated with deepTools (v3.1.3). Matrices were generated with computeMatrix followed by plotProfile or plotHeatmap as appropriate.

### Data quantification

Read quantification was carried out with featureCounts (v1.6.1) from the Subread suite (76). For repeated elements, files were obtained by extracting, in .bed file format, entries annotated as "SINE," "LINE," or "LTR" in the "RepClass" field from RepeatMasker.

### Analysis of splicing

Differential splicing analysis was done using rMATS (v4.1.0) (77) with the parameters: --libType fr-firststrand –novelSS. The occurrence of each type of splicing event was then counted.

### Reverse transcription (RT)–qPCR and 5′/3′ ratio analysis

Cells leftover after the monocyte isolation procedure, mostly T/B cells, were used for total RNA extraction by TRIzol as described above. RT–qPCR followed by DNase treatment was performed as previously described (13). The 5′/3′ ratio was calculated by comparing intronic RNA levels detected by RT–qPCR at each end of the genes under scrutiny. The 5′ primer sets were located ~1 kb downstream of the TSS, whereas the 3′ primer sets were near the TES. The ratio of 5′ intronic RNA levels to 3′ intronic RNA levels was first calculated for each patient. Then, the median of these ratios was determined. A list of primers is provided in Table S4.

### Cbx5 null mouse model

C57BL/6N-Atm1Brd Cbx5tm1a(EUCOMM)Wtsi/WtsiOrl strain inactivated for Cbx5 was received from the EUCOMM Consortium in the context of a Standard Material Transfer Agreement. It was transferred via the TAAM—CNRS Orléans for breeding at the Institut Pasteur—Paris. The experiments involving the mouse model were carried out in strict accordance with the approved research protocols and in compliance with all applicable institutional and regulatory guidelines (CETEA Institut Pasteur approval no. dap170030).

### EAE

EAE was carried out as previously described (78). Briefly, to prepare the MOG35-55 emulsion for immunization, we first calculated the total volume required and then prepared 1.5 to 2 times that amount to account for potential losses during the process. Each mouse received a subcutaneous injection of 200 $\mu$l of a mixture containing MOG35-55 peptide solution and Complete Freund's Adjuvant (CFA) in a 1:1 ratio. The MOG35-55 peptide solution was prepared by diluting lyophilized 200 $\mu$g of the peptide per mouse with ddH2O to achieve a final concentration of 2 mg/ml, which was then stored at −20°C. CFA was prepared by grinding 100 mg of desiccated *Mycobacterium tuberculosis* H37RA and adding 10 ml of Incomplete Freund's Adjuvant to create a 10 mg/ml CFA stock solution that could be stored at 4°C. Just before immunization, CFA was diluted with Incomplete Freund's Adjuvant to reach a final concentration of 2 mg/ml, with thorough mixing. The MOG35-55 and CFA were then mixed in a 1:1 ratio until the final concentration of 1 mg/ml was achieved. This emulsion, a critical step for immunization, was carefully prepared to ensure proper emulsification. After verification of its stability, the emulsion was drawn into syringes for subsequent use. In addition to the MOG35-55 emulsion, pertussis toxin was prepared and administered. For each mouse, 400 ng of pertussis toxin in 200 $\mu$l of PBS was injected intraperitoneally on the day of immunization and repeated 2 d later. Pertussis toxin was reconstituted by dissolving 50 $\mu$g in 500 $\mu$l of ddH2O to create a 100 $\mu$g/ml stock solution, which was stored at 4°C. To achieve the required concentration for immunization, the stock solution was diluted 1:50 with PBS. Finally, to ensure proper identification and monitoring, individual mice were marked, typically using color markings on the tail base. A second dose of pertussis toxin was administered on day 2 after the initial immunization to complete the immunization process. In this study, two independent sets of mice were used to examine the effects of the EAE protocol. Set 1: 10 WT mice, 10 *Cbx5*+/− mice, and 5 *Cbx5*−/− mice allowed observation of the evolution of EAE over 31 d. Set 2: 2 WT, 1 *Cbx5*+/−, and 3 *Cbx5*−/− mice allowed the RNA-seq approach. The latter set was interrupted at day 14.

## Data Availability

All data in the figures are available in the published article and in online supplemental material. Human monocyte and mouse Cbx5 KO RNA-seq data are available at the Gene Expression Omnibus respectively under accession no. GSE249613 and no. GSE249605.

## Supplementary Information

## Acknowledgements

The work was supported by a grant from REVIVE, an ANR "Laboratoire d'Excellence" program (2011–2021). T Christensen received funding from the Graduate School of Health, Aarhus University, the Danish MS Society, the Beckett Foundation, and the Dagmar Marshalls Foundation. Travel expenses of M Carstensen for the collaboration with C Muchardt were covered by the Knud Højgaards Foundation, the Augustinus Foundation, the Aage og Johanne Louis-Hansens Foundation, and the Oticon Foundation.

### Author Contributions

Y Porozhan: formal analysis, investigation, methodology, and writing—review and editing.
M Carstensen: conceptualization, funding acquisition, investigation, methodology, and writing—review and editing.
S Thouroude: investigation.
M Costallat: resources, data curation, and methodology.
C Rachez: formal analysis, investigation, methodology, and writing—review and editing.
E Batsché: formal analysis, methodology, and writing—review and editing.

T Petersen: conceptualization, supervision, funding acquisition, investigation, methodology, project administration, and writing—review and editing.

T Christensen: conceptualization, supervision, funding acquisition, investigation, methodology, project administration, and writing—review and editing.

C Muchardt: conceptualization, data curation, formal analysis, supervision, funding acquisition, investigation, methodology, project administration, and writing—original draft, review, and editing.

## Conflict of Interest Statement

The authors declare that they have no conflict of interest.

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
