## [Reviewer comments · Life Science Alliance]

Life Science Alliance

Defective Integrator activity shapes the transcriptome of patients with multiple sclerosis

Yevheniia Porozhan, Mikkel Carstensen, Sandrine Thouroude, Mickael Costallat, Christophe Rachez, Eric Batsché, Thor Petersen, Tove Christensen, and Christian Muchardt

DOI: <https://doi.org/10.26508/lsa.202402586>

Corresponding author(s): Christian Muchardt, Institut de Biologie Paris-Seine

Review Timeline:	Submission Date:	2024-01-11
	Editorial Decision:	2024-04-08
	Revision Received:	2024-06-28
	Editorial Decision:	2024-07-03
	Revision Received:	2024-07-10
	Accepted:	2024-07-10

Transaction Report:

April 8, 2024

Re: Life Science Alliance manuscript #LSA-2024-02586-T

Mr. Christian Muchardt
Institut de Biologie Paris-Seine
CNRS - UMR8256 - Biological Adaptation and Ageing
7-9, Quai Saint Bernard
Paris cedex 05 75252
France

Dear Dr. Muchardt,

Thank you for submitting your manuscript entitled "Defective Integrator activity shapes the transcriptome of patients with multiple sclerosis" to Life Science Alliance. The manuscript was assessed by expert reviewers, whose comments are appended to this letter. We invite you to submit a revised manuscript addressing the Reviewer comments.

Thank you for this interesting contribution to Life Science Alliance. We are looking forward to receiving your revised manuscript.

Sincerely,

B. MANUSCRIPT ORGANIZATION AND FORMATTING:

Reviewer #1 (Comments to the Authors (Required)):

The manuscript by Porozhan et al. entitled "Defective Integrator activity shapes the transcriptome of patients with multiple sclerosis" attempts at dissecting the role of the transcription factor CBX5 in the context of autoimmune demyelination. The authors identified by RNA-seq a sub-group of multiple sclerosis (MS) patients characterized by low levels of CBX5 in monocytes. Further analysis connected low CBX5 expression with extensive RNA dysmetabolism. Lastly, the authors were able to recapitulate many of the phenotypes observed in MS patients in CBX5 deficient mice with the MS model experimental autoimmune encephalomyelitis (EAE).

This work is relevant to the MS research field as it sheds light on a previously unknown molecular mechanism underlying MS etiology. The study does a good job in combining multiple approaches. The experiments appear well executed and the results generally support the conclusions. However, several technical and conceptual limitations decrease the robustness and impact of the work. For these reasons, the following points need to be carefully addressed before I can endorse the manuscript for publication:

-The observation of a subset of MS patients with low CBX5 levels needs to be independently replicated to exclude possible random effects due to the small sample size of the tested cohort and possible influence of confounding factors. For instance, the high-CBX5 group is enriched in CIS patients (likely treatment naïve) while the low-CBX5 group consists only in patients with clinically defined MS (likely under treatment).

-There is no rationale for the initial choice to profile monocytes in MS patients. Also, data on the purity of the isolated monocytes should be included as supplementary files.

-Although the symptomatic control group is an interesting choice, the initial RNA-seq screening requires an additional control group encompassing healthy subjects which will serve as real baseline for all the other experimental groups.

-The differences in the levels of CBX5 detected by RNA-seq need to be independently confirmed by qRT-PCR and western blotting.

-The plot of EAE clinical scores in Fig. 4B lacks the necessary statistics to confirm the significance of the differences between genotypes.

-The characterization of the EAE model is largely incomplete. The analysis of EAE scores should be complemented with histological evaluation of immune cell infiltration and demyelination in the spinal cord. In addition, the peripheral autoimmune response should be assessed in terms of pro-inflammatory vs tolerogenic mediators (Th1/Th17 T cells, cytokines and so on).

-It is not clear the number of mice that was used for the EAE experiment for each genotype. Same for the number of independent immunizations that were carried out. Given the high degree of variability of the model and the relatively small differences in clinical scores between genotypes, multiple immunizations are required to confirm the results.

-It is not clear the rationale for performing RNA-seq on CD4+ T cells in EAE mice while monocytes were investigated in MS patients. The analysis in EAE mice should be extended to monocytes for consistency.

-More than 200 loci have been found associated with MS risk in a recent GWAS (IMSGC, Science 2019). It would be relevant to test whether the low-CBX5 group carries any of those risk SNPs in the CBX5 locus.

Reviewer #2 (Comments to the Authors (Required)):

The authors characterize a subset of patients with reduced CBX5 expression using RNA-seq and find many gene expression changes, including impaired maturation of U snRNAs and enhancer RNAs. These RNAs are normally acted upon by the

Integrator complex and the authors confirm that the expression of several INTcom subunits are altered. Consistent with the idea of altered INTcom activity, genes regulated by Pol II pause release are also affected. The authors then inactivated Cbx5 in mouse and find this mirrors many of the transcriptional effects and makes the animal hypersensitive to EAE. Overall the work suggests the INTcom is an important player in preventing the transcriptional anomalies seen in MS. The manuscript is overall straightforward but I do have a number of suggestions that the authors should address to clarify data presentation.

- (1) Fig 2C: The authors have shown genome browser views for a few snRNA loci but they should look more systematically at all snRNA loci to determine if there is a general accumulation of 3' extensions at snRNAs across the genome.
- (2) Error bars are lacking on multiple graphs, including Fig 2B, 4H, and 5A.
- (3) Fig 5B: The authors write in the main text that the "read-count in the 3' regions was relatively higher" (Line 365) but this is not obvious from the figure shown.
- (4) There are multiple ways that Integrator has been proposed to act at protein-coding genes. The authors highlight one model that Pol II may have reduced competence for elongation when Integrator is not active, but other studies have shown that blocking Integrator can lead to large increases in mature mRNA levels (in contrast to what is written on Line 415). More inclusive language should be used throughout the manuscript.
- (5) Fig 2B, Fig 5A: Additional Integrator subunits have been identified beyond INTS1-12. These include INTS13, INTS14, INTS15 and PP2A subunits. The authors should analyze their expression levels as well.

Minor comments:

- (1) Fig 1B: Words in figure are small and hard to read.
- (2) Fig 1F: Please explain the figure labels more so that a reader can more easily understand how this figure shows an enrichment for Vitamin D regulated genes as described in the text.
- (3) Line 205: It could be interesting that INTS6 is up-regulated as recent work suggests INTS6 over expression can block Integrator activity. PMID: 37995689
- (4) Line 209: The authors should cite PMID:16239144 as this was the earliest report of Integrator acting on snRNAs.

Rebuttal

Reviewer #1:

The manuscript by Porozhan et al. entitled "Defective Integrator activity shapes the transcriptome of patients with multiple sclerosis" attempts at dissecting the role of the transcription factor CBX5 in the context of autoimmune demyelination. The authors identified by RNA-seq a sub-group of multiple sclerosis (MS) patients characterized by low levels of CBX5 in monocytes. Further analysis connected low CBX5 expression with extensive RNA dysmetabolism. Lastly, the authors were able to recapitulate many of the phenotypes observed in MS patients in CBX5 deficient mice with the MS model experimental autoimmune encephalomyelitis (EAE).

This work is relevant to the MS research field as it sheds light on a previously unknown molecular mechanism underlying MS etiology. The study does a good job in combining multiple approaches. The experiments appear well executed and the results generally support the conclusions.

We thank the Reviewer for recognizing the relevance of our work for MS research and for his positive comments on our approaches and execution.

However, several technical and conceptual limitations decrease the robustness and impact of the work. For these reasons, the following points need to be carefully addressed before I can endorse the manuscript for publication:

-The observation of a subset of MS patients with low CBX5 levels needs to be independently replicated to exclude possible random effects due to the small sample size of the tested cohort and possible influence of confounding factors. For instance, the high-CBX5 group is enriched in CIS patients (likely treatment naïve) while the low-CBX5 group consists only in patients with clinically defined MS (likely under treatment).

To increase the robustness of our findings, we now analyze a publicly available RNA-seq data set from MS patient T cells. This analysis confirms that between patients with respectively low and high CBX5 expression, the genes that are differentially expressed are those highly dependent on promoter escape for their regulation. These data are now presented in new Sup. Figures 2A-2D. Regarding the enrichment of the high-CBX5 and low-CBX5 groups in different categories of patients, we agree that the sample size is small and that the diagram Fig1D only indicates a trend. This is now clearly addressed in the "discussion" section.

-There is no rationale for the initial choice to profile monocytes in MS patients. Also, data on the purity of the isolated monocytes should be included as supplementary files.

Monocytes were chosen downstream of initial study by Carstensen et al, 2018 (Ref #18) showing increased levels of epitopes originating from endogenous retroviruses in monocytes from MS patients. These observations had suggested that monocytes would be a good tissue for the detection of unusual transcripts. This is now clearly stated in the revised version. Purity of the monocytes was verified by flow cytometry on patient #129, with the same approach as in Ref #18. This is now clearly stated in the "Material & Method" section, but unfortunately, we have not kept the .fcs files required to make the requested sup. figure.

-Although the symptomatic control group is an interesting choice, the initial RNA-seq screening requires an additional control group encompassing healthy subjects which will serve as real baseline for all the other experimental groups.

Concerning the use of symptomatic controls, we believe that we have lacked clarity in the initial version. SCs are healthy, and do not have any disease. They have experience paresthesia without change in sensation, blurry vision explained by a lack of adjustment in vision, dizziness due to increased muscle tension, reduced muscle strength due to decreased physical activity and condition, etc. Thus, they have neurological symptoms, but no objective clinical or paraclinical findings define a specific neurological disease. In combination with MRI showing normal or unspecific findings, the designation of SC is determined through exclusionary criteria and, clearly, does not represent early MS. Finally, because they have been through an MS diagnosis process, we have good evidence for a normal neurological examination and MRI of the central nervous system. Thus, we hope that the Reviewer will agree that SCs forms a better characterized and ultimately more homogeneous group of control than would even age- and gender-matched “healthy controls”.

-The differences in the levels of CBX5 detected by RNA-seq need to be independently confirmed by qRT-PCR and western blotting.

The entirety of the original monocyte preparations was lysed and all the yielded RNA was used for the preparation of the libraries used in the RNA-seq reactions. Thus, we can provide neither Westernblots, nor RT-qPCR reactions on these samples. Yet, we still had available the fractions complementary to the monocytes after the Miltenyi columns, enriched in T and B cells. PCR reactions on these samples has allowed to confirm the variations in CBX5 gene expression between the groups. As will be further described below, these leftover samples have also allowed us to document the defective pause-release in Lo-CBX5 patients, using cells other than monocytes. These data are now presented in new Figures 3E-3H and Sup. Figures 3B-3E.

-The plot of EAE clinical scores in Fig. 4B lacks the necessary statistics to confirm the significance of the differences between genotypes.

We have now clearly indicated that the error bars represent standard error of the mean (SEM), which was indeed missing. This was also corrected for panels Sup. 4A and Sup. 4B.

-The characterization of the EAE model is largely incomplete. The analysis of EAE scores should be complemented with histological evaluation of immune cell infiltration and demyelination in the spinal cord. In addition, the peripheral autoimmune response should be assessed in terms of pro-inflammatory vs tolerogenic mediators (Th1/Th17 T cells, cytokines and so on).

We recognize that a more thorough characterization of the EAE model is necessary for a comprehensive understanding of the impact of Cbx5 inactivation on mouse immunity. Unfortunately, none of the current co-authors have sufficient know-how in immunology to carry out this characterization. Thus, to carry out these experiments, we have teamed up with a group in Toulouse and are requesting funding for these experiments. However, these experiments will require importing the mice to Toulouse, which would not be compatible with the 3-month period allowed for the revision, and we hope the Reviewer agrees that this work should be included in a new manuscript with a different set of authors.

-It is not clear the number of mice that was used for the EAE experiment for each genotype. Same for the number of independent immunizations that were carried out. Given the high degree of variability of the model and the relatively small differences in clinical scores between genotypes, multiple immunizations are required to confirm the results.

EAE was carried out by immunizing 10 WT mice, 10 Cbx5+/- mice, 6 Cbx5 -/- mice (more difficult to breed). As one of Cbx5-/- mice died at day 12, presented results only include data from 5 of the KO mice. The RNA-seq data was carried out on a second set of mice including 2 WT, 1 Cbx5+/-, and 3 Cbx5-/- mice used as controls, and 3 WT and 3 Cbx5-/- mice exposed to an EAE protocol arrested at day 14. This is now clearly indicated in the Material and Method section of the manuscript. In addition, we have included in Sup. Figure 4B, the tacking of the body weight of the Cbx5-/- mice, which provides a quantitative complement to the clinical scores.

-It is not clear the rationale for performing RNA-seq on CD4+ T cells in EAE mice while monocytes were investigated in MS patients. The analysis in EAE mice should be extended to monocytes for consistency.

Repeating the RNA-seq analysis on monocytes from Cbx5 KO mice would not have been possible in the 3-month period allowed for the revision. However, to improve the consistency of the manuscript, we have (1) leveraged the complementary fractions from our monocyte purification, enriched in B and T cells, to examine manifestations of RNAPII pausing in non-monocyte cells at a couple of representative genes. This approach shows that patients accumulating reads on the 5' region of genes in monocytes also yield a greater RT-PCR signal on this region (as compared to more 3' regions) in the PBMCs leftover from the monocyte purification. These data are now presented in new Figure 3E-3H and Sup. Figure 3C-3E. In addition, we have examined publicly available transcriptomic data from MS patient CD4+ T cells (Kim et al. 2020 – New Ref #22). Our analysis shows that by segregating patients based on CBX5 expression levels, genes with significantly altered expression between high and low CBX5 groups are enriched in NELFE targets, similar to our observations in monocytes. These findings strongly suggest that in T cells, as in monocytes, reduced levels of CBX5 are primarily associated with the deregulation of genes highly dependent on promoter escape for their regulation. These results are now presented in Sup. Figures 2A-2D.

-More than 200 loci have been found associated with MS risk in a recent GWAS (IMSGC, Science 2019). It would be relevant to test whether the low-CBX5 group carries any of those risk SNPs in the CBX5 locus.

The IMSGC study identifies a risk SNP in INTS8. Therefore, the association of INTS8 with an increased risk of MS can now be supported by two different references (Ref #26, and #27 - IMSGC), and we thank the reviewer for highlighting this. Conversely, CBX5 is, to our knowledge, not associated with an increased risk of MS, including findings from the IMSGC study. It is possible that alterations in the CBX5 gene may be lethal in humans. This point is now addressed in the discussion section.

Reviewer #2:

The authors characterize a subset of patients with reduced CBX5 expression using RNA-seq and find many gene expression changes, including impaired maturation of U snRNAs and enhancer RNAs. These RNAs are normally acted upon by the Integrator complex and the authors confirm that the expression of several INTcom subunits are altered. Consistent with the idea of altered INTcom activity, genes regulated by Pol II pause release are also affected. The authors then inactivated Cbx5 in mouse and find this mirrors many of the transcriptional effects and makes the animal hypersensitive to EAE. Overall the work suggests the INTcom is an important player in preventing the transcriptional anomalies seen in MS. The manuscript is overall straightforward but I do have a number of suggestions that the authors should address to clarify data presentation.

(1) Fig 2C: The authors have shown genome browser views for a few snRNA loci but they should look more systematically at all snRNA loci to determine if there is a general accumulation of 3' extensions at snRNAs across the genome.

We have identified a total of 17 loci where the snRNAs are expressed and sufficiently distant from other transcribed genes to permit the detection of its imperfect maturation. These data are now compiled in a new Sup. Figure 2F.

(2) Error bars are lacking on multiple graphs, including Fig 2B, 4H, and 5A.

Error bars have been added to these graphs.

(3) Fig 5B: The authors write in the main text that the "read-count in the 3' regions was relatively higher" (Line 365) but this is not obvious from the figure shown.

We agree that the increased transcription downstream of the snRNAs was difficult to see in the example provided. To improve this and also to provide more systematic evidence for transcriptional defects possibly linked to decreased Integrator activity, we have (1) changed Figure 5B to select a new locus where the non-matured forms of snRNAs are easier to see.

(2) Included a new Sup. Figure 5A showing that in average, accumulation of reads over snRNA genes is reduced in Cbx5 mutant mice, an observation consistent with poor maturation of these ncRNAs. Finally, (3) we provide a systematic analysis of histone genes, that like snRNAs, are dependent on the Integrator complex for their maturation. Specifically, we now show that 62 out of the 72 annotated histone genes in the mouse genome are down-regulated in the Cbx5 mutants. These data are presented in the new Sup. Fig 5B.

(4) There are multiple ways that Integrator has been proposed to act at protein-coding genes. The authors highlight one model that Pol II may have reduced competence for elongation when Integrator is not active, but other studies have shown that blocking Integrator can lead to large increases in mature mRNA levels (in contrast to what is written on Line 415). More inclusive language should be used throughout the manuscript.

This section has been rephrased to emphasize that we are only describing one of multiple possible activities of INTcom.

(5) Fig 2B, Fig 5A: Additional Integrator subunits have been identified beyond INTS1-12. These include INTS13, INTS14, INTS15 and PP2A subunits. The authors should analyze their expression levels as well.

These subunits have been included in Fig 2B (human) and 5A (mouse).

Minor comments:

(1) Fig 1B: Words in figure are small and hard to read.

The font has been adjusted to match other text in the Figure (Arial 10).

(2) Fig 1F: Please explain the figure labels more so that a reader can more easily understand how this figure shows an enrichment for Vitamin D regulated genes as described in the text. We now explain in the Figure legend that VDR stands for vitamin D receptor.

(3) Line 205: It could be interesting that INTS6 is up-regulated as recent work suggests INTS6 over expression can block Integrator activity. PMID: 37995689

This reference is now cited (Reference #40) and we thank the reviewer for the suggestion.

(4) Line 209: The authors should cite PMID:16239144 as this was the earliest report of Integrator acting on snRNAs.

This reference is now cited (Reference #28).

July 3, 2024

RE: Life Science Alliance Manuscript #LSA-2024-02586-TR

Mr. Christian Muchardt
Institut de Biologie Paris-Seine
CNRS - UMR8256 - Biological Adaptation and Ageing
7-9, Quai Saint Bernard
Paris cedex 05 75252
France

Dear Dr. Muchardt,

Thank you for submitting your revised manuscript entitled "Defective Integrator activity shapes the transcriptome of patients with multiple sclerosis". We would be happy to publish your paper in Life Science Alliance pending final revisions necessary to meet our formatting guidelines.

- please be sure that the authorship listing and order is correct
- please upload all figure files as individual ones, including the supplementary figure files
- please add a Category for your manuscript in our system
- please add the Twitter handle of your host institute/organization as well as your own or/and one of the authors in our system
- please label the Introduction section.
- please add an Author Contributions section to your main manuscript text
- please add a Conflict of Interest statement to your main manuscript text
- please separate the Figure legends and Supplemental Figure legends into separate sections
- we encourage you to revise the figure legend for Figure S4 such that the figure panels are introduced in alphabetical order
- please add callouts for Figures 3L; S3D, E and S5B, C to your main manuscript text

LSA now encourages authors to provide a 30-60 second video where the study is briefly explained. We will use these videos on social media to promote the published paper and the presenting author (for examples, see <https://docs.google.com/document/d/1-UWCfbE4pGcDdcgzcmiuJl2XMBJnxKYeqRvLLrLS08s/edit?usp=sharing>). Corresponding or first-authors are welcome to submit the video. Please submit only one video per manuscript. The video can be emailed to contact@life-science-alliance.org

A. FINAL FILES:

B. MANUSCRIPT ORGANIZATION AND FORMATTING:

Sincerely,

July 10, 2024

RE: Life Science Alliance Manuscript #LSA-2024-02586-TRR

Mr. Christian Muchardt
Institut de Biologie Paris-Seine
CNRS - UMR8256 - Biological Adaptation and Ageing
7-9, Quai Saint Bernard
Paris cedex 05 75252
France

Dear Dr. Muchardt,

Thank you for submitting your Research Article entitled "Defective Integrator activity shapes the transcriptome of patients with multiple sclerosis". It is a pleasure to let you know that your manuscript is now accepted for publication in Life Science Alliance. Congratulations on this interesting work.

DISTRIBUTION OF MATERIALS:

Again, congratulations on a very nice paper. I hope you found the review process to be constructive and are pleased with how the manuscript was handled editorially. We look forward to future exciting submissions from your lab.

Sincerely,
